# The Everlasting Database:
# Statistical Validity at a Fair Price

**Blake Woodworth**
Toyota Technological
Institute at Chicago

**Vitaly Feldman**
Google

**Saharon Rosset**
Tel Aviv University

**Nathan Srebro**
Toyota Technological
Institute at Chicago

## Abstract

The problem of handling adaptivity in data analysis, intentional or not, permeates a variety of fields, including test-set overfitting in ML challenges and the accumulation of invalid scientific discoveries. We propose a mechanism for answering an arbitrarily long sequence of potentially adaptive statistical queries, by charging a price for each query and using the proceeds to collect additional samples. Crucially, we guarantee statistical validity without any assumptions on how the queries are generated. We also ensure with high probability that the cost for $M$ non-adaptive queries is $O(\log M)$, while the cost to a potentially adaptive user who makes $M$ queries that do not depend on any others is $O(\sqrt{M})$.

## 1 Introduction

Consider the problem of running a server that provides the test loss of a model on held out data, e.g. for evaluation in a machine learning challenge. We would like to ensure that all test losses returned by the server are accurate estimates of the true generalization error of the predictors.

Returning the empirical error on held out test data would initially be a good estimate of the generalization error. However, an analyst can use the empirical errors to adjust their model and improve their performance on the test data. In fact, with a number of queries only linear in the amount of test data, one can easily create a predictor that completely overfits, having empirical error on the test data that is artificially small [5, 12]. Even without such intentional overfitting, sequential querying can lead to unintentional adaptation since analysts are biased toward tweaks that lead to improved test errors.

If the queries were non-adaptive, i.e. the sequence of predictors is not influenced by previous test results, then we could handle a much larger number of queries before overfitting–a number exponential in the size of the dataset. Nevertheless, the test set will eventually be "used up" and estimates of the test error (specifically those of the best performers) might be over-optimistic.

A similar situation arises in other contexts such as validating potential scientific discoveries. One can evaluate potential discoveries using set aside validation data, but if analyses are refined adaptively based on the results, one may again overfit the validation data and arrive at false discoveries [14, 17].

One way to ensure the validity of answers in the face of adaptive querying is to collect all queries before giving any answers, and answer them all at once, e.g. at the end of a competition. However, analysts typically want more immediate feedback, both for ML challenges and in scientific research. Additionally, if we want to answer more queries later, ensuring statistical validity would require collecting a whole new dataset. This might be unnecessarily expensive if few or none of the queries are in fact adaptive. It also raises the question of who should bear the cost of collecting new data.

Alternatively, we could try to limit the number or frequency of queries from each user, forbid adaptive querying, or assume users work independently of each other, remaining oblivious to other users' queries and answers. However, it is nearly impossible to enforce such restrictions. Determined users

can avoid querying restrictions by creating spurious user accounts and working in groups; there is no feasible way to check if queries are chosen adaptively; and information can leak between analysts, intentionally or not, e.g. through explicit collaboration or published results.

In this paper, we address the fundamental challenge of providing statistically valid answers to an arbitrarily long sequence of potentially adaptive queries. We assume that it is possible to collect additional samples from the same data distribution at a fixed cost per sample. To pay for new samples, users of the database will be charged for their queries. We propose a mechanism, EVERLASTINGVALIDATION, that guarantees "everlasting" statistical validity and maintains the following properties:

**Validity** Without any assumptions about the users, and even with arbitrary adaptivity, with high probability, all answers ever returned by the database are accurate.

**Self-Sustainability** The database collects enough revenue to purchase as many new samples as necessary in perpetuity, and can answer an *unlimited* number of queries.

**Cost for Non-Adaptive Users** With high probability, a user making $M$ non-adaptive queries will pay at most $O(\log M)$, so the average cost per query decreases as $\tilde{O}(1/M)$.

**Cost for Autonomous Users** With high probability, a user (or group of users) making $M$ potentially adaptive queries that depend on each other arbitrarily, but not on any queries made by others, will pay at most $\tilde{O}(\sqrt{M})$, so the average cost per query decreases as $\tilde{O}(1/\sqrt{M})$.

We emphasize that the database mechanism needs no notion of "user" or "account" when answering the queries; it does not need to know which "user" made which query; and most of all, it does not need to know whether a query was made adaptively or not. Rather, the cost guarantees hold for any collection of queries that are either non-adaptive or autonomous in the sense described above–a "user" could thus refer to a single individual, or if an analyst uses answers from another person's queries, we can consider them together as an "autonomous user" and get cost guarantees based on their combined number of queries. The database's cost guarantees are nearly optimal; the cost to non-adaptive users and the cost to autonomous users cannot be improved (beyond log-factors) while still maintaining validity and sustainability (Section 5).

As is indicated by the guarantees above, using the mechanism adaptively may be far more expensive than using it non-adaptively. We view this as a positive feature. Although we cannot enforce non-adaptivity, and it is sometimes unreasonable to expect that analysts are entirely non-adaptive, we intend the mechanism to be used for *validation*. That is, analysts should do their discovery, training, tuning, development, and adaptive data analysis on unrestricted "training" or "discovery" datasets, and only use the protected database when they wish to receive a stamp of approval on their model, predictor, or discovery. Instead of trying to police or forbid adaptivity, we discourage it with pricing, but in a way that is essentially guaranteed not to affect non-adaptive users. Further, users will need to pay a high price only when their queries explicitly cause overfitting, so only adaptivity that is harmful to statistical validity will be penalized.

**Relationship to prior work** Our work is inspired by a number of mechanisms for dealing with potentially adaptive queries that have been proposed and analyzed using techniques from differential privacy and information theory. These mechanisms handle only a pre-determined number of queries using a fixed dataset. We use techniques developed in this literature, in particular addition of noise to ensure that a quadratically larger number of adaptive queries can be answered in the worst case [6, 10]. Our main innovations over this prior work are the self-sustaining nature of the database, as opposed to handling only a pre-determined number of queries of each type, and also the per-query pricing scheme that places the cost burden on the adaptive users. To ensure that the cost burden on non-adaptive users does not grow by more than a constant factor, we need to adapt existing algorithms.

LADDER [5] and SHAKYLADDER [16] are mechanisms tailored to maintaining a ML competition leaderboard. These algorithms reveal the answer to a user's query for the error of their model only if it is significantly lower than the error of the previous best submission from the user. While these mechanisms can handle an exponential number of arbitrarily adaptive submissions, each user will receive answers to a relatively small number of queries. Our setting is more suitable for the case where we want to validate the errors of all submissions or for scientific discovery where there is more then one discovery to be made.

A separate line of work in the statistics literature on "Quality Preserving Databases" (Aharoni and Rosset [2] and references therein) has suggested schemes for databases that maintain everlasting validity, while charging for use. The fundamental difference from our work is that these schemes

do not account for adaptivity and thus are limited to non-adaptive querying. A second difference is that they focus on hypothesis testing for scientific discovery, with pricing schemes that depend on considerations of statistical power, which are not part of our framework. We further compare with existing methods at the end of Section 4.

## 2   Model formulation

We consider a setting in which a database curator has access to samples from some unknown distribution $\mathcal{D}$ over a sample space $\mathcal{X}$. Multiple analysts submit a sequence of statistical queries $q_i : \mathcal{X} \to [0, 1]$, the database responds with answers $a_i \in \mathbb{R}$, and the goal is to ensure that with high probability, all answers satisfy $|a_i - \mathbb{E}_{x \sim \mathcal{D}}[q_i(x)]| \leq \tau$ for some fixed accuracy parameter $\tau$. In a prediction validation application, each query would measure the expected loss of a particular model, while in scientific applications a single query might measure the value of some phenomenon of interest, or compare it to a "null" reference. We denote $\mathcal{Q}$ the set of all possible queries, i.e. measurable functions $q : \mathcal{X} \to [0, 1]$, and use the shorthand $\mathbb{E}[q] = \mathbb{E}_{x \sim \mathcal{D}}[q(x)]$ to denote the mean value (desired answer) for each query. Given a data sample $S \sim \mathcal{D}^n$, we use $\mathcal{E}_S[q] = \frac{1}{|S|} \sum_{x \in S} q(x)$ as shorthand for the empirical mean of $q$ on $S$.

In our framework, the database can, at any time, acquire new samples from $\mathcal{D}$ at some fixed cost per sample, e.g. by running more experiments or paying workers to label more data. To answer a given query, the database can use the samples it has already purchased in any way it chooses, and the database is allowed to charge analysts for their queries in order to purchase additional samples. The price $p_i$ of query $q_i$ may be determined by the database after it receives query $q_i$, allowing the database to charge more for queries that force it to collect more data.

We do not assume the queries are chosen in advance, and instead allow the sequence of queries to depend adaptively on past answers. More formally, we define a "querying rule" $R_i : (\mathcal{Q}, \mathbb{R}, \mathbb{R})^{i-1} \mapsto \mathcal{Q}$ as a randomized mapping from the history of all previously made queries and their answers and prices to the statistical query to be made next:

$$q_i = R_i \left( (q_1, a_1, p_1), (q_2, a_2, p_2), \ldots, (q_{i-1}, a_{i-1}, p_{i-1}) \right).$$

The interaction of users with the database can then be modeled as a sequence of querying rules $\{R_i\}_{i \in \mathbb{N}}$. The combination of the data distribution, database mechanism, and sequence of querying rules together define a joint distribution over queries, answers, and prices $\{Q_i, A_i, P_i\}_{i \in \mathbb{N}}$. All our results will hold for any data distribution and any querying sequence, with high probability over $\{Q_i, A_i, P_i\}_{i \in \mathbb{N}}$.

We think of the query sequence as representing a combination of queries from multiple users, but the database itself is unaware of the identity or behavior of the users. Our validity guarantees do not assume any particular user structure, nor any constraints on the interactions of the different users. Thus, the guarantees are always valid regardless of what a "user" means, how "users" are allowed to collaborate, how many "users" there are, or how many queries each "user" makes—the guarantees simply hold for any (arbitrarily adaptive) querying sequence.

However, our cost guarantees will, and must, refer to analysts (or perhaps groups of analysts) behaving in specific ways. In particular, we define a **non-adaptive user** as a subsequence $\{u_j\}_{j \in [M]}$ consisting of queries which do not depend on *any* of the history, i.e. $R_{u_j}$ is a fixed (pre-determined) distribution over queries, so $Q_{u_j}$ is independent of all of the history. We further define an **autonomous user** of the database as a subsequence $\{u_j\}_{j \in [M]}$ of the querying rules that depend only on the history *within the subsequence*, i.e.

$$R_{u_j} \left( (q_1, a_1, p_1), \ldots, (q_{(u_j - 1)}, a_{(u_j - 1)}, p_{(u_j - 1)}) \right) =$$
$$R_{u_j} \left( (q_{u_1}, a_{u_1}, p_{u_1}), \ldots, (q_{u_{(j-1)}}, a_{u_{(j-1)}}, p_{u_{(j-1)}}) \right).$$

That is, $Q_{u_j}$ is independent of the overall past history given the past history pertaining to the autonomous user. The "cost to a user" is the total price paid for queries in the subsequence $\{u_j\}$: $\sum_{j=1}^{M} p_{u_j}$.

# 3  VALIDATIONROUND

Our mechanism for providing "everlasting" validity guarantees is based on a query answering mechanism which we call VALIDATIONROUND. It uses $n$ samples from $\mathcal{D}$ in order to answer $\exp(\Omega(n))$ non-adaptive and at least $\tilde{\Omega}(n^2)$ adaptive statistical queries (and potentially many more). Our analysis is based on ideas developed in the context of adaptive data analysis [10] and relies on techniques from differential privacy [9]. Differential privacy is a strong stability property of randomized algorithms that operate on a dataset. Composition properties of differential privacy imply that this form of stability holds even when the same dataset is used by multiple algorithms that can depend on the outputs of preceding algorithms. Most importantly, differential privacy implies generalization with high probability [4, 10].

VALIDATIONROUND splits its data into two sets $S$ and $T$. Upon receiving each query, it first checks whether the answers on these datasets approximately agree. If so, the query has almost certainly not overfit to the data, and the algorithm simply returns the empirical mean of the query on $S$ plus additional random noise. We show that the addition of noise ensures that the algorithm, as a function from the data sample $S$ to an answer, satisfies differential privacy. This can be leveraged to show that any query which depends on a limited number of previous queries will have an empirical mean on $S$ that is close to the true expectation. This ensures that VALIDATIONROUND can accurately answer a large number of queries, while allowing some (unknown) subset of the queries to be adaptive.

VALIDATIONROUND uses truncated Gaussian noise $\xi \sim \mathcal{N}(0, \sigma^2, [-\gamma, \gamma])$, i.e. Gaussian noise $Z \sim \mathcal{N}(0, \sigma^2)$ conditioned on the event $|Z| \leq \gamma$. Its density $f_\xi(x) \propto \exp\left(-\frac{x^2}{2\sigma^2}\right) \mathbb{1}_{|x| \leq \gamma}$.

---

**Algorithm 1** VALIDATIONROUND$(\tau, \beta, n, S, T)$

---

1: Set $I(\tau, \beta, n) = \frac{\beta}{4} \exp\left(\frac{n\tau^2}{8}\right)$, $\sigma^2 = \frac{\tau^2}{32 \ln(8n^2/\beta)}$
2: **for** each query $q_1, q_2, \ldots$ **do**
3:    **if** $|\mathcal{E}_S[q_i] - \mathcal{E}_T[q_i]| \leq \frac{\tau}{2}$ and $i \leq I(\tau, \beta, n)$ **then**
4:       Draw truncated Gaussian $\xi_i \sim \mathcal{N}(0, \sigma^2, [-\frac{\tau}{4}, \frac{\tau}{4}])$
5:       **Output:** $a_i = \mathcal{E}_S[q_i] + \xi_i$
6:    **else**
7:       **Halt** $(\eta = i)$

---

Here, $\eta$ is the index of the query that causes the algorithm to halt. If $\eta \leq I(\tau, \beta, n)$, the maximum allowed number of answers, we say that VALIDATIONROUND halted "prematurely." The following three lemmas characterize the behavior of VALIDATIONROUND.

**Lemma 1.** *For any $\tau$, $\beta$, and $n$, for any sequence of querying rules (with arbitrary adaptivity) and any probability distribution $\mathcal{D}$, the answers provided by* VALIDATIONROUND$(\tau, \beta, n, S, T)$ *satisfy*

$$\mathbb{P}\left[\forall_{i < \eta} \left| A_i - \mathbb{E}_{x \sim \mathcal{D}}[Q_i(x)] \right| \leq \tau\right] \geq 1 - \frac{\beta}{2},$$

*where the probability is taken over the randomness in the draw of datasets $S$ and $T$ from $\mathcal{D}^n$, the querying rules, and* VALIDATIONROUND.

**Lemma 2.** *For any $\tau$, $\beta$, and $n$, any sequence of querying rules, and any non-adaptive user $\{u_j\}_{j \in [M]}$ interacting with* VALIDATIONROUND$(\tau, \beta, n, S, T)$, $\mathbb{P}\left[\eta \leq I(\tau, \beta, n) \wedge \eta \in \{u_j\}_{j \in [M]}\right] \leq \beta$.

**Lemma 3.** *For any $\tau$, $\beta$, and $n$, any sequence of querying rules, and any autonomous user $\{u_j\}_{j \in [M]}$ interacting with* VALIDATIONROUND$(\tau, \beta, n, S, T)$, *if $\sigma^2 = \frac{\tau^2}{32 \ln(8n^2/\beta)}$ and $M \leq \frac{n^2\tau^4}{175760 \ln^2(8n^2/\beta)}$ then $\mathbb{P}\left[\eta \leq I(\tau, \beta, n) \wedge \eta \in \{u_j\}_{j \in [M]}\right] \leq \beta$.*

Lemma 1 indicates that all returned answers are accurate with high probability, regardless of adaptivity. The proof involves showing that $\mathcal{E}_T[q_i]$ is close to $\mathbb{E}[q_i]$ for each query, so any query that is answered must be accurate since $|\mathcal{E}_S[q_i] - \mathcal{E}_T[q_i]|$ and $|\xi|$ are small. Lemma 2 indicates that with high probability, non-adaptive queries never cause a premature halt, which is a simple application of

Hoeffding's inequality. Finally, Lemma 3 shows that with high probability, an autonomous user who makes $\tilde{\mathcal{O}}(n^2)$ queries will not cause a premature halt. This requires showing that $\mathcal{E}_S[q_i]$ is close to $\mathbb{E}[q_i]$ despite the potential adaptivity.

The proof of Lemma 3 uses existing results from adaptive data analysis together with a simple argument that noise truncation does not significantly affect the results. For reference, the results we cite are included in Appendix E. While using Gaussian noise to answer queries is mentioned in other work, we are not aware of an explicit analysis, so we analyze the method here. To simplify parts of the derivation, we rely on the notion of concentrated differential privacy, which is particularly well suited for analysis of composition with Gaussian noise addition [6]. Lemmas 1-3 are proven in Appendix A.

## 4 EVERLASTINGVALIDATION and pricing

VALIDATIONROUND uses a fixed number, $n$, of samples and with high probability returns accurate answers for at least $\exp(\Omega(n))$ non-adaptive queries and $\tilde{\Omega}(n^2)$ adaptive queries. In order to handle infinitely many queries, we chain together multiple instances of VALIDATIONROUND. We start with an initial dataset, answer queries using VALIDATIONROUND using that data until it halts. At this point, we buy more data and repeat. The used-up data can be released to the public as a "training set," which can be used with no restriction without affecting any guarantees.

---

**Algorithm 2** EVERLASTINGVALIDATION$(\tau, \beta)$

1: Require initial budget $\Gamma = 36 \ln(8/\beta)/\tau^2$
2: $N_0 = \frac{\Gamma}{2}, \beta_0 = \frac{\beta}{2}, t = 0, i = 0$
3: Buy datasets $S_0, T_0 \sim \mathcal{D}^{N_0}$
4: **loop**
5:     Pass $q_i$ to VALIDATIONROUND$(\tau, \beta_t, N_t, S_t, T_t)$
6:     **if** VALIDATIONROUND does not halt **then**
7:         **Output:** $a_i$
8:         Charge $\frac{96}{\tau^2} \cdot \frac{1}{i}$, move on to $i = i + 1$
9:     **else**
10:        Charge $6N_t$ minus current capital
11:        $N_{t+1} = 3N_t, \beta_{t+1} = \frac{1}{2}\beta_t, t = t + 1$
12:        Buy datasets $S_t, T_t \sim \mathcal{D}^{N_t}$
13:        Restart loop with same $i$

---

The key ingredient is a pricing system with which we can always afford new data when an instance of VALIDATIONROUND halts. Our method has two price types: a low price, which is charged for all queries and decreases like $1/i$; and a high price, which is charged for any query that causes an instance of VALIDATIONROUND to halt prematurely, which may grow with the size of the current dataset. EVERLASTINGVALIDATION$(\tau, \beta)$ guarantees the following:

**Theorem 1** (Validity). *For any sequence of querying rules (with arbitrary adaptivity),* EVERLASTINGVALIDATION *will provide answers such that*

$$\mathbb{P}\left[\forall_{i \in \mathbb{N}} \left| A_i - \mathbb{E}_{x \sim \mathcal{D}}[Q_i(x)] \right| \leq \tau \right] \geq 1 - \frac{\beta}{2}$$

*Proof.* Consider the sequence of query rules that are answered by the $t^{\text{th}}$ instantiation of the VALIDATIONROUND mechanism. By Lemma 1, for any sequence of querying rules, with probability $1 - \frac{\beta_t}{2}$, all of the answers during round $t$ are answered accurately. By a union bound over all rounds, all answers in all rounds are accurate with probability at least $1 - \sum_{t=0}^{\infty} \beta_t/2 = 1 - \beta/2$.   □

**Theorem 2** (Sustainability). *For any sequence of queries, the revenue collected can pay for all samples ever needed by* EVERLASTINGVALIDATION, *excluding the initial budget of* $36 \ln(8/\beta)/\tau^2$.

*Proof.* When VALIDATIONROUND halts, we charge exactly enough for the next $S_t, T_t$ (line 10).   □

**Lemma 4.** *If $N_0 \geq 18\ln(2)/\tau^2$ and $I(\tau, \beta_t, N_t) = (\beta_t/4)\exp\left(N_t\tau^2/8\right)$ queries are answered during round $t$, then at least $6N_t$ revenue is collected.*

The proof of Lemma 4 involves a straightforward computation. We find an upper bound, $B_T$, on the number of queries made before round $T$ begins and then lower bound the revenue collected in round $T$ with $\sum_i \frac{96}{\tau^2(B_T+i)}$. We defer the details to Appendix B.

**Theorem 3** (Cost for non-adaptive users). *For any sequence of querying rules and any non-adaptive user indexed by $\{u_j\}_{j\in[M]}$, the cost to the user satisfies*

$$\mathbb{P}\left[\sum_{j\in[M]} P_{u_j} \leq \frac{96}{\tau^2}\left(1 + \ln(M)\right)\right] \geq 1 - \beta.$$

*Proof.* By Lemma 4, if a round $t$ ends after $I(\tau, \beta_t, N_t)$ queries are answered, then the total revenue collected from queries in that round is at least $6N_t$, so the "high price" at the end of the round is $0$. Consequently, a query $q_{u_j}$ from the non-adaptive user costs the low price $96/(\tau^2 u_j)$ unless it causes an instantiation of VALIDATIONROUND to halt prematurely. By Lemma 2 and a union bound, this never occurs in any round with probability at least $1 - \sum_{t=0}^{\infty}\beta_t = 1 - \beta$, and the cost to the user is

$$\sum_{j\in[M]} p_{u_j} = \sum_{j\in[M]} \frac{96}{\tau^2 u_j} \leq \sum_{j\in[M]} \frac{96}{\tau^2 i} \leq \frac{96}{\tau^2}\left(1 + \ln(M)\right). \qquad \square$$

**Theorem 4** (Cost for adaptive users). *For any sequence of querying rules and any autonomous user indexed by $\{u_j\}_{j\in[M]}$, there is a fixed constant $c_0$ such that the cost to the user satisfies*

$$\mathbb{P}\left[\sum_{j\in[M]} P_{u_j} \leq c_0 \cdot \frac{\sqrt{M}\ln^2\left(M/\beta\right)}{\tau^2}\right] \geq 1 - \beta.$$

*Proof.* Ideally, none of the $M$ queries causes a premature halt, and the total cost is at most $\frac{96}{\tau^2}\left(1 + \ln(M)\right)$, but the adaptive user may cause rounds to end prematurely and pay up to $6N_t$. However, by Lemma 3, with probability $1 - \beta_t$ if one of the adaptive user's queries causes a round $t$ to end prematurely, then the amount of data, $N_t$, and the number of the user's queries answered in that round, $M_t$, must satisfy

$$M_t \geq \frac{N_t^2 \tau^4}{175760 \ln^2\left(8N_t^2/\beta_t\right)}. \tag{1}$$

Given $M$, there is a largest $t$ for which this is possible since $N_t = 3^t N_0$ and $\beta_t = 2^{-t}\beta_0$. That is,

$$\frac{3^{2t} N_0^2 \tau^4}{175760 \ln\left(18^t \cdot 8N_0^2/\beta_0\right)} \leq M$$

which implies $t_{\max} \leq \frac{1}{2}\ln\left(24\sqrt{M}\ln\left(144N_0/\beta_0\right)\right)$. Let $\mathcal{T}$ be the set of rounds in which the adaptive user pays the high $6N_t$ price, then with probability at least $1 - \sum_{t\in\mathcal{T}}\beta_t \geq 1 - \beta$, inequality (1) holds for all $t \in \mathcal{T}$. In this case, the total cost to the adaptive user is no more than

$$\sum_{t\in\mathcal{T}} 6N_t \leq t_{\max}\frac{2520\sqrt{M}\ln\left(8M^2/\beta_{t_{\max}}\right)}{\tau^2} \leq \frac{1890\sqrt{M}\ln^2\left(16M^2/\beta\right)}{\tau^2}. \qquad \square$$

**Relationship to prior work on adaptive data analysis**   We handle adaptivity using ideas developed in recent work on adaptive data analysis. In this line of work, all queries are typically assumed to be adaptively chosen and the overall number of queries known in advance. For completeness, we briefly describe several algorithms that have been developed in this context and compare them with our algorithm. Dwork et al. [10] analyze an algorithm that adds Laplace or Gaussian noise to the empirical mean in order to answer $M$ adaptive queries using $\tilde{O}(\sqrt{M})$ samples—a method that forms the basis of VALIDATIONROUND. However, adding untruncated Laplace or Gaussian noise to exponentially many non-adaptive queries would likely cause large errors when the variance is large enough to ensure that the sample mean is accurate. We use truncated Gaussian noise instead and show that it does not substantially affect the analysis for autonomous queries.

THRESHOLDOUT [11] answers verification queries in which the user submits both a query and an estimate of the answer. The algorithm uses $n = \tilde{O}(\sqrt{M} \cdot \log I)$ samples to answer $I$ queries of which at most $M$ estimates are far from correct. Similar to our use of the second dataset $T$, this algorithm can be used to detect overfitting and answer adaptive queries (this is the basis of the EFFECTIVEROUNDS algorithm [10]). However, in our application this algorithm would have sample complexity of $n = \tilde{O}(\sqrt{M} \cdot \log I)$, for $M$ autonomous queries in $T$ total queries. Consequently, direct use of this mechanism would result in a pricing for non-adaptive users that depends on the number of queries by autonomous users. This is in contrast to $n = \tilde{O}(\sqrt{M} + \log T)$ samples that suffice for VALIDATIONROUND, where the improvement relies on our definition of autonomy and truncation of the noise variables.

## 5 Optimality

One might ask if it is possible to devise a mechanism with similar properties but lower costs. We argue that the prices set by EVERLASTINGVALIDATION are near optimal. The total cost to a non-adaptive user who makes $M$ queries is $O(\log M/\tau^2)$. Even if we knew in advance that we would receive only $M$ non-adaptive queries, we would still need $\Omega(\log M/\tau^2)$ samples to answer all of them accurately with high probability. Thus, our price for non-adaptive queries is optimal up to constant factors.

It is also known that answering a sequence of $M$ adaptively chosen queries with accuracy $\tau$ requires $\tilde{\Omega}(\sqrt{M}/\tau)$ samples [15, 19]. Hence, the cost to a possibly adaptive autonomous user is nearly optimal in its dependence on $M$ (up to log factors). One natural concern is that our guarantee in this case is only for the amortized (or total) cost, and not on the cost of each individual query. Indeed, although the *average* cost of adaptive queries decreases as $\tilde{O}(1/\sqrt{M})$, the *maximal* cost of a single query might increase as $\tilde{O}(\sqrt{M})$. A natural question is whether the maximum price can be reduced, to spread the high price over more queries.

Finally, an individual who queries our mechanism with $M$ entirely non-adaptive queries will only pay $\log M$ in the worst case; generally, they will benefit from the economies of scale associated with collecting more and more data. For instance, if there are $K$ users each making $M$ non-adaptive queries, then the total cost of all $KM$ queries will be $\log KM$ so the average cost to each user is only $\log(KM)/K \ll \log M$.

## 6 An Alternative Approach: EVERLASTINGTO

The EVERLASTINGVALIDATION mechanism provides cost guarantees that are, in certain ways, nearly optimal. The two main shortcomings are that (1) the price is guaranteed only for non-adaptive or autonomous users–not arbitrary adaptive ones and (2) the cost of an individual adaptive query cannot be upper bounded. One might also ask if inventing VALIDATIONROUND was necessary in the first place. Another mechanism, THRESHOLDOUT [11], is already well-suited to the setting of mixed adaptive and non-adaptive queries and it gives accuracy guarantees for quadratically many arbitrary adaptive queries or exponentially many non-adaptive queries. Perhaps using THRESHOLDOUT instead would be better? We will now describe an alternative mechanism, EVERLASTINGTO, which allows us to provide price guarantees for individual queries, including arbitrarily adaptive ones, but with an exponential increase in the cost for both non-adaptive and adaptive queries.

The EVERLASTINGTO mechanism is very similar to EVERLASTINGVALIDATION, except it uses THRESHOLDOUT in the place of VALIDATIONROUND. In each round, the algorithm determines an overfitting budget, $B_t$, and a maximum number of queries, $M_t$, as a function of the tradeoff parameter $p$. It then answers queries using THRESHOLDOUT, charging a high price $2N_{t+1}/B_t$ for queries that fail the overfitting check, and charging a low price $2N_{t+1}/M_t$ for all of the other queries. Once THRESHOLDOUT cannot answer more queries, the mechanism buys more data, reinitializes THRESHOLDOUT, and continues as before.

We analyze EVERLASTINGTO in Appendix D. Theorems 6-9 closely parallel the guarantees of EVERLASTINGVALIDATION and establish the following for any $\tau, \beta \in (0, 1)$ and any $p \in (0, \frac{2}{3})$: **Validity**: with high probability, for any sequence of querying rules, all answers provided by EVERLASTINGTO are $\tau$-accurate. **Sustainability**: EVERLASTINGTO charges high enough prices to be able to afford new samples as needed, excluding the initial budget. **Cost**: with high probability, any

---

**Algorithm 3** EVERLASTINGTO$(\tau, \beta, p)$

---

1: Require sufficiently initial budget $n = n(\tau, \beta, p)$
2: $\forall t$ set $N_t = ne^t$, $\beta_t = \frac{(e-1)\beta}{e}e^{-t}$, $B_t = \tilde{\Theta}\left(\frac{\tau^4 N_t^{2-2p}}{\ln 1/\beta_t}\right)$, $M_t = \frac{\beta_t}{4}\exp\left(2N_t^p\right)$
3: **for** $t = 0, 1, \ldots$ **do**
4:     Purchase datasets $S_t, T_t \sim \mathcal{D}^{N_t}$ and initialize THRESHOLDOUT$(S_t, T_t, B_t, \beta_t)$.
5:     **while** THRESHOLDOUT$(S_t, T_t, B_t, \beta_t)$ has not halted **do**
6:         Accept query $q$
7:         $(a, o) = $ THRESHOLDOUT$(S_t, T_t, B_t, \beta_t)(q)$
8:         **Output**: $a$
9:         **if** $o = \perp$ **then**
10:             **Charge**: $\frac{2N_{t+1}}{M_t}$
11:         **else**
12:             **Charge**: $\frac{2N_{t+1}}{B_t}$

---

$M$ non-adaptive queries and any $B$ adaptive queries cost at most $O\left(\ln^{1/p}(M) + B^{\frac{1}{2-3p}}\right)$ (ignoring the dependence on $\tau, \beta$).

Unlike EVERLASTINGVALIDATION, which prioritized charging as little as possible for non-adaptive queries, EVERLASTINGTO increases the $O(\log M)$ cost to $O(\text{polylog }M)$ in order to bound the price of arbitrary adaptive queries. The parameter $p$ allows the database manager to control the tradeoff; for $p$ near zero, the cost of $B$ adaptive queries is roughly the optimal $O(\sqrt{B})$, but non-adaptive queries are extremely expensive. On the other side, for $p$ near $2/3$, the cost of adaptive queries becomes very high, but the cost of non-adaptive queries is relatively small, although it does not approach optimality.

Further details of the mechanism are contained in Appendix D. We also provide a tighter analysis of the THRESHOLDOUT algorithm which guarantees accurate answers using a substantially smaller amount of data in Appendix C. This analysis allows us to reduce the exponent in EVERLASTINGTO's cost guarantee for non-adaptive queries.

## 7 Potential applications

In the ML challenge scenario, validation results are often displayed on a scoreboard. Although it is often assumed that scoreboards cannot be used for extensive adaptation, it appears that such adaptations have played roles in determining the outcome of various well known competitions, including the Netflix challenge, where the final test set performance was significantly worse than performance on the leaderboard data set. EVERLASTINGVALIDATION would guarantee that test errors returned by the validation database are accurate, regardless of adaptation, collusion, the number of queries made by each user, or other intentional or unintentional dependencies. We do charge a price per-validation, but as long as users are non-adaptive, the price is very small. Adaptive users, on the other hand, pay what is required in order to ensure validity (which could be a lot). Nevertheless, even if a wealthy user could afford paying the higher cost of adaptive queries, she would still not be able to "cheat" and overfit the scoreboard set, and a poor user could still afford the quickly diminishing costs of validating non-adaptive queries.

Another feature of our mechanism is that once a round $t$ is over, we can safely release the datasets $S_t$ and $T_t$ to the public as unrestricted training data. This way, poor analysts also benefit from adaptive queries made by others, as all data is eventually released, and at any given time, a substantial fraction of all the data ever collected is public. Also, the ratio of public data to validation data can easily be adjusted by slightly amending the pricing.

In the context of scientific discovery, one use case is very similar to the ML competition. Scientists can search for interesting phenomena using unprotected data, and then re-evaluate "interesting" discoveries with the database mechanism in order to get an accurate and almost-unbiased estimate of the true value. This could be useful, for example, in building prediction models for scientific phenomena such as genetic risk of disease, which often involve complex modeling [7].

However, most scientific research is done in the context of hypothesis testing rather than estimation. Declarations of discoveries like the Higgs boson [1] and genetic associations of disease [8] are based

on performing a potentially large number of hypothesis tests and identifying statistically significant discoveries while controlling for multiplicity. Because of the complexity of the discovery process, it is often quite difficult to properly control for all potential tests, causing many difficulties, the most well known of which is the problem of publication bias (cf. "Why Most Published Research Findings are False" [17]). An alternative, approach that has gained popularity in recent years, is requiring replication of any declared discoveries on new and independent data [3]. Because the new data is used only for replication, it is much easier to control multiplicity and false discovery concerns.

Our everlasting database can be useful in both the discovery and replication phases. We now briefly explain how its validity guarantees can be used for multiplicity control in testing. Assume we have a collection of hypothesis tests on functionals of $\mathcal{D}$ with null hypotheses: $H_{0i} : \mathbb{E}[q_i] = e_{0i}$. We employ our scheme to obtain estimates $A_i$ of $\mathbb{E}[q_i]$. Setting $\alpha = \beta/2$, Theorem (1) guarantees: $\sum_i \mathbb{P}_{H_{0i}} [\max_i |A_i - e_{0i}| > \tau] \leq \alpha$, meaning that for any combination of true nulls, the rejection policy *reject if* $|A_i - e_{0i}| > \tau$ makes no false rejections with probability at least $1 - \alpha$, thus controlling the family-wise error rate (FWER) at level $\alpha$. This is easily used in the replication phase, where an entire community (say, type-I diabetes researchers) could share a single replication server using the everlasting database scheme in order to to guarantee validity. It could also be used in the discovery phase for analyses that can be described through a set of measurements and tests of the form above.

## 8 Conclusion and extensions

Our primary contribution is in designing a database mechanism that brings together two important properties that have not been previously combined: everlasting validity and robustness to adaptivity. Furthermore, we do so in an asymptotically efficient manner that guarantees that non-adaptive queries are inexpensive with high probability, and that the potentially high cost of handling adaptivity only falls upon truly adaptive users. Currently, there are large constants in the cost guarantees, but these are pessimistic and can likely be reduced with a tighter analysis and more refined pricing scheme. We believe that with some improvements, our scheme can form the basis of practical implementations for use in ML competitions and scientific discovery. Also, our cost guarantees themselves are worst-case and only guarantee a low price to entirely non-adaptive users. It would be useful to investigate experimentally how much users would actually end up being charged under "typical use," especially users who are only "slightly adaptive." However, there is no established framework for understanding what would constitute "typical" or "slightly adaptive" usage of a statistical query answering mechanism, so more work is needed before such experiments would be insightful.

Our mechanism can be improved in several ways. It only provides answers at a fixed, additive $\tau$, and only answers statistical queries, however these issues have been already addressed in the adaptive data analysis literature. E.g. arbitrary low-sensitivity queries can be handled without any modification to the algorithm, and arbitrary real-valued queries can be answered with the error proportional to their standard deviation (instead of $1/\sqrt{n}$ as in our analysis) [13]. These approaches can be combined with our algorithms but we restrict our attention to the basic case since our focus is different.

Finally, one potentially objectionable element of our approach is that it discards samples at the end of each round (although these samples are not wasted since they become part of the public dataset). An alternative approach is to add the new samples to the dataset as they can be purchased. While this might be a more practical approach, existing analysis techniques that are based on differential privacy do not appear to suffice for dealing with such mechanisms. Developing more flexible analysis techniques for this purpose is another natural direction for future work.

**Acknowledgements**  BW is supported the NSF Graduate Research Fellowship under award 1754881.

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
