[Supplementary Material]

# A  Proofs from Section 3

**Lemma 5.** *For any $\tau$, $\beta$, $n$, and any sequence of querying rules (with arbitrary adaptivity) interacting with* VALIDATIONROUND$(\tau, \beta, n, S, T)$

$$\mathbb{P}\left[\forall_{i<\eta} \left|\mathcal{E}_T\left[Q_i\right] - \mathbb{E}_{x\sim\mathcal{D}}\left[Q_i(x)\right]\right| \leq \frac{\tau}{4}\right] \geq 1 - \frac{\beta}{2}.$$

*Proof.* Consider any sequence of querying rules (with arbitrary adaptivity). The interaction between the query rules and VALIDATIONROUND$(\tau, \beta, n, S, T)$ together determines a joint distribution over statistical queries, answers, and prices $(Q_1, A_1, P_1), ..., (Q_{\eta-1}, A_{\eta-1}, P_{\eta-1})$.

Consider also the interaction of the same sequence of querying rules with an alternative algorithm, which always returns $\mathcal{E}_S\left[q_i\right] + \xi_i$ (i.e. it ignores the if-statement in VALIDATIONROUND). This generates an infinite sequence of queries, answers, and prices $(Q_1', A_1', P_1'), (Q_2', A_2', P_2'), ....$ Now, we retroactively check the condition in the if-statement for each of the queries to calculate what $\eta$ should be, and take the length $\eta - 1$ prefix of the $(Q_i', A_i', P_i')$. This sequence has exactly the same distribution as the sequence generated by VALIDATIONROUND, and each $Q_i'$ was chosen independently of $T$ by construction. Since $Q_i' \sim Q_i$ has outputs bounded in $[0, 1]$, we can apply Hoeffding's inequality:

$$\mathbb{P}\left[\left|\mathcal{E}_T\left[Q_i\right] - \mathbb{E}_{x\sim\mathcal{D}}\left[Q_i(x)\right]\right| > \frac{\tau}{4}\right] \leq 2\exp\left(-\frac{n\tau^2}{8}\right).$$

At most $I(\tau, \beta, n) = \frac{\beta}{4}\exp\left(\frac{n\tau^2}{8}\right)$ queries are answered by the mechanism, so a union bound completes the proof. □

**Lemma 1.** *For any $\tau$, $\beta$, and $n$, for any sequence of querying rules (with arbitrary adaptivity) and any probability distribution $\mathcal{D}$, the answers provided by* VALIDATIONROUND$(\tau, \beta, n, S, T)$ *satisfy*

$$\mathbb{P}\left[\forall_{i<\eta} \left|A_i - \mathbb{E}_{x\sim\mathcal{D}}\left[Q_i(x)\right]\right| \leq \tau\right] \geq 1 - \frac{\beta}{2},$$

*where the probability is taken over the randomness in the draw of datasets $S$ and $T$ from $\mathcal{D}^n$, the querying rules, and* VALIDATIONROUND.

*Proof.* A query is not answered unless $\left|\mathcal{E}_S\left[q_i\right] - \mathcal{E}_T\left[q_i\right]\right| \leq \frac{\tau}{2}$, so $\forall i < \eta$

$$\left|a_i - \mathbb{E}\left[q_i\right]\right| \leq \left|\xi_i\right| + \left|\mathcal{E}_S\left[q_i\right] - \mathcal{E}_T\left[q_i\right]\right| + \left|\mathcal{E}_T\left[q_i\right] - \mathbb{E}\left[q_i\right]\right| \leq \tau/4 + \tau/2 + \left|\mathcal{E}_T\left[q_i\right] - \mathbb{E}\left[q_i\right]\right|.$$

By Lemma 5, with probabilty $1 - \frac{\beta}{2}$ the final term is at most $\tau/4$ simultaneously for all $i < \eta$. □

**Lemma 2.** *For any $\tau$, $\beta$, and $n$, any sequence of querying rules, and any non-adaptive user $\{u_j\}_{j\in[M]}$ interacting with* VALIDATIONROUND$(\tau, \beta, n, S, T)$, $\mathbb{P}\left[\eta \leq I(\tau, \beta, n) \wedge \eta \in \{u_j\}_{j\in[M]}\right] \leq \beta$.

*Proof.* Since the non-adaptive user's querying rules ignore all of the history, they are each chosen independently of $S$. By Hoeffding's inequality

$$\mathbb{P}\left[\left|\mathcal{E}_S\left[Q_{u_j}\right] - \mathbb{E}_{x\sim\mathcal{D}}\left[Q_{u_j}(x)\right]\right| > \frac{\tau}{4}\right] \leq 2\exp\left(-\frac{n\tau^2}{8}\right)$$

and similarly for $T$. If both $\eta \leq I(\tau, \beta, n)$ and $\eta = u_j$, then the algorithm halted upon receiving query $q_{u_j}$ because its empirical means on $S$ and $T$ were too dissimilar and *not* because it had already answered its maximum allotment of queries. Therefore,

$$\mathbb{P}\left[\eta \leq I(\tau, \beta, n) \wedge \eta = u_j\right] = \mathbb{P}\left[\left|\mathcal{E}_S\left[Q_{u_j}\right] - \mathcal{E}_T\left[Q_{u_j}\right]\right| > \frac{\tau}{2}\right] \leq 4\exp\left(-\frac{n\tau^2}{8}\right).$$

At most $I(\tau, \beta, n) = \frac{\beta}{4}\exp\left(\frac{n\tau^2}{8}\right)$ queries are answered by the mechanism, so a union bound completes the proof. □

**Lemma 6.** *For any $\tau$, $\beta$, $n$, any sequence of query rules, and any possibly adaptive autonomous user $\{u_j\}_{j\in[M]}$, if $\sigma^2 = \frac{\tau^2}{32\ln(8n^2/\beta)}$ and $M \leq \frac{n^2\tau^4}{175760\ln^2(8n^2/\beta)}$ then*

$$\mathbb{P}\left[\forall_{j\in[M]} \; \left|\mathcal{E}_S\left[Q_{u_j}\right] - \mathbb{E}_{x\sim\mathcal{D}}\left[Q_{u_j}(x)\right]\right| \leq \frac{\tau}{4}\right] \geq 1 - \frac{\beta}{2}.$$

*Proof.* Consider a slightly modified version of VALIDATIONROUND, where Gaussian noise $z_i \sim \mathcal{N}(0,\sigma^2)$ is added instead of truncated Gaussian noise $\xi_i$. Until this modified algorithm halts, all of the answers it provides are released according to the Gaussian mechanism on $S$, which satisfies $\frac{1}{2n^2\sigma^2}$-zCDP by Proposition 1.6 in [6]. We can view $Q_{u_j} = R_{u_j}((q_{u_1}, a_{u_1}, p_{u_1})...,(q_{u_{j-1}}, a_{u_{j-1}}, p_{u_{j-1}}))$ as an (at most) $M$-fold composition of $\frac{1}{2n^2\sigma^2}$-zCDP mechanisms, which satisfies $\frac{M}{2n^2\sigma^2}$-zCDP by Lemma 1.7 in [6]. Finally, Proposition 1.3 in [6] shows us how to convert this concentrated differential privacy guarantee to a regular differential privacy guarantee. In particular, $q_{u_j}$ is generated under

$$\left(\frac{M}{2n^2\sigma^2} + 2\sqrt{\frac{M}{2n^2\sigma^2}\ln(1/\delta)}, \; \delta\right)\text{-DP} \quad \forall \delta > 0.$$

Specifically, when $\sigma^2$, $\delta$ and $M$ satisfy:

$$\sigma^2 = \frac{\tau^2}{32\ln(8n^2/\beta)}$$

$$\delta = \frac{\beta}{8n^2} = \frac{\beta}{\frac{n^2\tau}{13\ln(104/\tau)}} \cdot \frac{\tau}{104\ln(104/\tau)}$$

$$M \leq \frac{n^2\tau^4}{175760\ln^2(8n^2/\beta)}.$$

then $q_{i_j}$ is generated by a $\left(\frac{\tau}{52}, \delta\right)$-differentially private mechanism. Therefore, by Theorem 8 in [11] (cf. [4, 18])

$$\mathbb{P}\left[\left|\mathcal{E}_S\left[q_{u_j}\right] - \mathbb{E}\left[q_{u_j}\right]\right| > \frac{\tau}{4}\right] \leq \frac{\beta}{\frac{n^2\tau}{13\ln(104/\tau)}} \ll \frac{\beta}{4M}.$$

Furthermore, for $z_i \sim \mathcal{N}(0,\sigma^2)$ $\mathbb{P}\left[|z_i| \geq \tau/4\right] \leq \beta/(4n^2) \leq \beta/(4M)$. Therefore, the total variation distance between $\xi_{u_j} \sim \mathcal{N}\left(0,\sigma^2,[-\tau/4,\tau/4]\right)$ and $z_{u_j} \sim \mathcal{N}(0,\sigma^2)$ is $\Delta(\xi_{u_j}, z_{u_j}) = \mathbb{P}\left[z_{u_j} \notin [-\tau/4,\tau/4]\right] \leq \frac{\beta}{4M}$. Consider two random vectors $Z$ and $\xi$, the first of which has independent $\mathcal{N}(0,\sigma^2)$ distributed coordinates, and the second of which has coordinates $\xi_{u_j} \sim \mathcal{N}\left(0,\sigma^2,[-\tau/4,\tau/4]\right)$ for $j \in [M]$ and $\xi_i = Z_i$ for all of the $i \notin \{u_j\}$. The total variation distance between these vectors is then at most $\Delta(\xi, Z) \leq M\Delta(\xi_{u_j}, z_{u_j}) \leq \frac{\beta}{4}$.

Now, for the given sequence of querying rules, $S$, and $T$, view VALIDATION-ROUND as a function of the random noise which is added into the answers. Then $\Delta(\text{VALIDATIONROUND}(\xi), \text{VALIDATIONROUND}(Z)) \leq \Delta(\xi, Z) \leq \frac{\beta}{4}$ too. Above, we showed that with probability $1 - \beta/4$ the user's interaction with VALIDATIONROUND$(Z)$ has the property that

$$\mathbb{P}\left[\exists_{j\in[M]} \; \left|\mathcal{E}_S\left[q_{u_j}\right] - \mathbb{E}\left[q_{u_j}\right]\right| > \frac{\tau}{4}\right] \leq \frac{\beta}{4}.$$

So their interaction with VALIDATIONROUND$(\xi)$ satisfies

$$\mathbb{P}\left[\exists_{j\in[M]} \; \left|\mathcal{E}_S\left[q_{u_j}\right] - \mathbb{E}\left[q_{u_j}\right]\right| > \frac{\tau}{4}\right] \leq \frac{\beta}{2}.$$

Since this statement only depends on the indices of $\xi$ in $\{u_j\}_{j\in[M]}$, we can replace all of the remaining indices with truncated Gaussians and maintain this property, which recovers VALIDATIONROUND. $\square$

**Lemma 3.** *For any $\tau$, $\beta$, and $n$, any sequence of querying rules, and any autonomous user $\{u_j\}_{j\in[M]}$ interacting with VALIDATIONROUND$(\tau, \beta, n, S, T)$, if $\sigma^2 = \frac{\tau^2}{32\ln(8n^2/\beta)}$ and $M \leq \frac{n^2\tau^4}{175760\ln^2(8n^2/\beta)}$ then $\mathbb{P}\left[\eta \leq I(\tau, \beta, n) \wedge \eta \in \{u_j\}_{j\in[M]}\right] \leq \beta$.*

*Proof of Lemma 3.* Consider a query $q_{u_j}$ made by the autonomous user. Lemma 5 guarantees that

$$\mathbb{P}\left[\forall_{j \in [M]} \ \left|\mathcal{E}_T\left[q_{u_j}\right] - \mathbb{E}\left[q_{u_j}\right]\right| \leq \frac{\tau}{4}\right] \geq 1 - \frac{\beta}{2}.$$

By Lemma 6, with the hypothesized $\sigma^2$ and $M$

$$\mathbb{P}\left[\forall_{j \in [M]} \ \left|\mathcal{E}_S\left[q_{u_j}\right] - \mathbb{E}\left[q_{u_j}\right]\right| \leq \frac{\tau}{4}\right] \geq 1 - \frac{\beta}{2}.$$

If both $\eta \leq I(\tau, \beta, n)$ and $\eta \in \{u_j\}_{j \in [M]}$, then the algorithm halted upon receiving a query $q_{u_j}$ because its empirical means on $S$ and $T$ were too dissimilar and *not* because it had already answered its maximum allotment of queries:

$$\mathbb{P}\left[\eta \leq I(\tau, \beta, n) \wedge \eta \in \{u_j\}_{j \in [M]}\right]$$
$$= \mathbb{P}\left[\exists_{j \in [M]} \ \left|\mathcal{E}_S\left[q_{u_j}\right] - \mathcal{E}_T\left[q_{u_j}\right]\right| > \frac{\tau}{2}\right] \leq \beta. \qquad \square$$

## B Proofs of Lemma 4

**Lemma 4.** *If $N_0 \geq 18 \ln(2)/\tau^2$ and $I(\tau, \beta_t, N_t) = (\beta_t/4) \exp\left(N_t \tau^2/8\right)$ queries are answered during round $t$, then at least $6N_t$ revenue is collected.*

*Proof.* The revenue collected in round $t$ via the low price $\frac{96}{\tau^2 i}$ depends on how many queries are answered both in and before round $t$. The maximum number of queries answered in a round is $I_t = I(\tau, \beta_t, N_t) = (\beta_t/4) \exp\left(N_t \tau^2/8\right)$ (this is enforced by VALIDATIONROUND). Let $B_T$ be the total number of queries made before the beginning of round $T$, then

$$B_T \leq \sum_{t=0}^{T-1} I_t = \sum_{t=0}^{T-1} \frac{\beta_t}{4} \exp\left(\frac{N_t \tau^2}{8}\right)$$
$$\leq \frac{\beta_0}{4} \exp\left(\sum_{t=0}^{T-1} \frac{\tau^2}{8} 3^t N_0 - t \ln 2\right)$$
$$\leq (\beta_T/4) \exp\left(N_T \tau^2/16\right).$$

The first inequality holds because every exponent in the sum is at least $\ln(2)$ by our choice of $N_0$ and for any $x, y \geq \ln 2$, $e^{x+y} \geq 2 \max\left(e^x, e^y\right) \geq e^x + e^y$. The second inequality holds since $N_0 > \frac{18 \ln 2}{\tau^2}$ implies $-T^2 + 3T - N_0 \tau^2/(8 \ln 2) \leq 0$. So, if $I_T$ queries are answered during round $T$, the revenue collected is at least

$$\sum_{i=1}^{I_T} \frac{96}{\tau^2(B_T + i)} \geq \frac{96}{\tau^2} \left(\ln\left(B_T + I_T\right) - \ln\left(B_T\right)\right)$$
$$\geq \frac{96}{\tau^2} \ln\left(1 + \frac{(\beta_T/4) \exp\left(N_T \tau^2/8\right)}{(\beta_T/4) \exp\left(N_T \tau^2/16\right)}\right)$$
$$\geq 6N_T \qquad \square$$

## C Tighter THRESHOLDOUT Analysis

In this section, we provide a tighter analysis of the THRESHOLDOUT algorithm [11]. In particular, previous analysis showed a sample complexity for answering $m$ queries with an overfitting budget of $B$ of $\tilde{O}(\sqrt{B} \ln^{1.5} m)$ whereas we prove a bound like $\tilde{O}(\sqrt{B} \ln m)$. The improvement has important consequences for our application of THRESHOLDOUT to the everlasting database setting. We make the improvement by applying the "monitor technique" of Bassily et al. [4].

**Lemma 7** (Lemma 23 [11]). THRESHOLDOUT *satisfies* $\left(\frac{2B}{\sigma n}, 0\right)$*-differential privacy and also* $\left(\frac{\sqrt{32B \ln(2/\delta)}}{\sigma n}, \delta\right)$*-differential privacy for any $\delta > 0$.*

**Algorithm 4** THRESHOLDOUT$(S, T, \tau, \beta, \zeta, B, \sigma)$

1: Sample $\rho \sim \text{Laplace}\,(2\sigma)$
2: **for** each query $q$ **do**
3:    **if** $B < 1$ **then**
4:       **HALT**
5:    **else**
6:       Sample $\lambda \sim \text{Laplace}\,(4\sigma)$
7:       **if** $|\mathcal{E}_S\,[q] - \mathcal{E}_T\,[q]| > \zeta + \rho + \lambda$ **then**
8:          Sample $\xi \sim \text{Laplace}\,(\sigma)$, $\rho \sim \text{Laplace}\,(2\sigma)$
9:          $B \leftarrow B - 1$
10:         **Output**: $(\mathcal{E}_T\,[q] + \xi, \top)$
11:       **else**
12:         **Output**: $(\mathcal{E}_S\,[q], \bot)$

**Lemma 8** (Corollary 7 [11]). *Let $\mathcal{A}$ be an algorithm that outputs a statistical query $q$. Let $S$ be a random dataset chosen according to distribution $\mathcal{D}^n$ and let $q = \mathcal{A}(S)$. If $\mathcal{A}$ is $\epsilon$-differentially private then*

$$\mathbb{P}\left[|\mathcal{E}_S\,[q] - \mathbb{E}\,[q]| \geq \epsilon\right] \leq 6\exp\left(-n\epsilon^2\right)$$

**Lemma 9** (Theorem 8 [11]). *Let $\mathcal{A}$ be an $(\epsilon, \delta)$-differentially private algorithm that outputs a statistical query. For dataset $S$ drawn from $\mathcal{D}^n$, we let $q = \mathcal{A}(S)$. Then for $n \geq \frac{2\ln(8/\delta)}{\epsilon^2}$,*

$$\mathbb{P}\left[|\mathcal{E}_S\,[q] - \mathbb{E}\,[q]| > 13\epsilon\right] \leq \frac{2\delta}{\epsilon}\ln\left(\frac{2}{\epsilon}\right)$$

**Theorem 5** (cf. Theorem 25 [11]). *Let $\beta, \tau > 0$ and $m \geq B > 0$. Set $\zeta = \frac{3\tau}{4}$ and $\sigma = \frac{\tau}{48\ln(4m/\beta)}$. Let $S, T$ denote datasets of size $n$ drawn i.i.d. from a distribution $\mathcal{D}$. Consider an analyst that is given access to $S$ and adaptively chooses functions $q_1, \ldots, q_m$ while interacting with THRESHOLDOUT which is given datasets $S, T$ and values $\sigma, B, \zeta$. For every $i \in [m]$ let $(a_i, o_i)$ denote the answer of THRESHOLDOUT on query $q_i$. Then whenever*

$$n \geq \min\left\{\mathcal{O}\left(\frac{B\ln\left(\frac{m}{\beta}\right)}{\tau^2}\right), \mathcal{O}\left(\frac{\ln\left(\frac{m}{\beta}\right)\sqrt{B\ln\left(\frac{\ln(1/\tau)}{\beta\tau}\right)}}{\tau^2}\right)\right\}$$

*with probability at least $1 - \beta$, for all $i$ before THRESHOLDOUT halts $|a_i - \mathbb{E}\,[q_i]| \leq \tau$ and $o_i = \top \implies q_i$ is an adaptive query.*

*Proof.* Consider the following post-processing of the output of THRESHOLDOUT: look through the sequence of queries and answers $(q_1, a_1), \ldots, (q_{\text{HALT}}, a_{\text{HALT}})$ and output $q^*, a^* = \arg\max_{q,a} |a - \mathbb{E}\,[q]|$. Since this procedure does not use the datasets $S, T$ and since THRESHOLDOUT computes the sequence of queries and answers in a differentially private manner, it means that $q^*, a^*$ are also released under differential privacy. So by Lemma 7, $q^*$ is released simultaneously under

$$\left(\frac{2B}{\sigma n}, 0\right)\text{-differential privacy} \qquad \text{and} \qquad \left(\frac{\sqrt{32B\ln\,(2/\delta)}}{\sigma n}, \delta\right)\text{-differential privacy} \quad (2)$$

With our choice of $\sigma$, in the case that $n \geq \frac{768B\ln\left(\frac{4m}{\beta}\right)}{\tau^2}$ then, using the pure differential privacy guarantee we have $\frac{2B}{\sigma n} \leq \frac{\tau}{8}$ so by Lemma 8

$$\mathbb{P}\left[|\mathcal{E}_T\,[q^*] - \mathbb{E}\,[q^*]| > \frac{\tau}{8}\right] \leq \frac{\beta}{4} \quad (3)$$

Alternatively, in the case that

$$n \geq \max\left\{\frac{9984\ln\left(\frac{4m}{\beta}\right)\sqrt{32B\ln\left(\frac{1664\ln\left(\frac{208}{\tau}\right)}{\beta\tau}\right)}}{\tau^2}, \frac{21632\ln\left(\frac{6656\ln\left(\frac{208}{\tau}\right)}{\beta\tau}\right)}{\tau^2}\right\}$$

then, choosing $\delta = \frac{\beta\tau}{832\ln\left(\frac{208}{\tau}\right)}$, under the approximate differential privacy guarantee we have

$$\left(\frac{\sqrt{32B\ln(2/\delta)}}{\sigma n}, \delta\right) \preceq \left(\frac{\tau}{104}, \frac{\beta\tau}{832\ln\left(\frac{208}{\tau}\right)}\right) \qquad (4)$$

so by Lemma 9

$$\mathbb{P}\left[|\mathcal{E}_T[q^*] - \mathbb{E}[q^*]| > \frac{\tau}{8}\right] \leq \frac{\beta}{4} \qquad (5)$$

Therefore, in either case $\mathbb{P}\left[|\mathcal{E}_T[q^*] - \mathbb{E}[q^*]| > \frac{\tau}{8}\right] \leq \frac{\beta}{4}$.

Next, we note that the random variable $\lambda$ is sampled at most $m$ times, and the random variables $\rho$ and $\xi$ are sampled at most $B$ times. Consequently,

$$\mathbb{P}\left[\exists i \ |\lambda_i| > \frac{\tau}{12}\right] \leq m \cdot \mathbb{P}\left[\left|\text{Laplace}\left(\frac{\tau}{12\ln(4m/\beta)}\right)\right| > \frac{\tau}{12}\right] \leq \frac{\beta}{4} \qquad (6)$$

$$\mathbb{P}\left[\exists i \ |\rho_i| > \frac{\tau}{24}\right] \leq B \cdot \mathbb{P}\left[\left|\text{Laplace}\left(\frac{\tau}{24\ln(4m/\beta)}\right)\right| > \frac{\tau}{24}\right] \leq \frac{\beta}{4} \qquad (7)$$

$$\mathbb{P}\left[\exists i \ |\xi_i| > \frac{7\tau}{8}\right] \leq B \cdot \mathbb{P}\left[\left|\text{Laplace}\left(\frac{\tau}{48\ln(4m/\beta)}\right)\right| > \frac{7\tau}{8}\right] \leq \frac{\beta}{8} \qquad (8)$$

For the rest of the proof, we condition on the events $|\mathcal{E}_T[q^*] - \mathbb{E}[q^*]| \leq \frac{\tau}{8}$ and $\forall i \ |\lambda_i| < \frac{\tau}{12}$, $|\rho_i| < \frac{\tau}{24}$, and $|\xi_i| < \frac{7\tau}{8}$. This event happens with probability $1 - \frac{7\beta}{8}$.

Consider two alternatives: either $a^* = \mathcal{E}_T[q^*] + \xi^*$ or $a^* = \mathcal{E}_S[q^*]$. In the first case,

$$|a^* - \mathbb{E}[q^*]| \leq |a^* - \mathcal{E}_T[q^*]| + |\xi^*| \leq \frac{\tau}{8} + \frac{7\tau}{8} = \tau \qquad (9)$$

In the second case, we also have that $|\mathcal{E}_S[q^*] - \mathcal{E}_T[q^*]| < \zeta + \rho^* + \lambda^*$, so

$$|a^* - \mathbb{E}[q^*]| \leq |\mathcal{E}_S[q^*] - \mathcal{E}_T[q^*]| + |\mathcal{E}_T[q^*] - \mathbb{E}[q^*]| \leq \zeta + |\rho^*| + |\lambda^*| + \frac{\tau}{8} \leq \frac{3\tau}{4} + \frac{\tau}{24} + \frac{\tau}{12} + \frac{\tau}{8} = \tau \qquad (10)$$

Therefore, for all queries before THRESHOLDOUT halts, $|a_i - \mathbb{E}[q_i]| \leq \tau$.

Next, observe that if $q$ is a non-adaptive query, then

$$\mathbb{P}\left[|\mathcal{E}_S[q] - \mathbb{E}[q]| > \frac{\tau}{4}\right] = \mathbb{P}\left[|\mathcal{E}_T[q] - \mathbb{E}[q]| > \frac{\tau}{4}\right] \leq 2\exp\left(-\frac{\tau^2 n}{8}\right) \leq 2\exp\left(50\ln\left(\frac{\beta}{4m}\right)\right) \leq \frac{2\beta}{m \cdot 4^{50}} \qquad (11)$$

Therefore, with probability at least $1 - \frac{\beta}{8}$, for all non-adaptive queries $|\mathcal{E}_S[q] - \mathcal{E}_T[q]| \leq \frac{\tau}{2}$. Furthermore,

$$\zeta + \rho + \lambda \geq \frac{3\tau}{4} - \frac{\tau}{24} - \frac{\tau}{12} = \frac{5\tau}{8} \qquad (12)$$

Thus, for all non-adaptive queries $|\mathcal{E}_S[q_i] - \mathcal{E}_T[q_i]| \leq \zeta + \rho_i + \lambda_i$, so $o_i = \perp$. □

## D   Guarantees of EVERLASTINGTO

**Theorem 6.** *[Validity] For any $\tau, \beta, p \in (0,1)$ and for a sufficiently large initial budget and for any sequence of queries,* EVERLASTINGTO *returns answers such that*
$$\mathbb{P}\left[\exists i \ |a_i - \mathbb{E}[q_i]| > \tau\right] < \beta$$

*Proof.* In round $t$, the algorithm uses an instance of THRESHOLDOUT with $N_t$ samples for the datasets $S_t$ and $T_t$, so to answer $M_t$ total queries of which at most $B_t$ overfit we need both

$$N_t = ne^t \geq \frac{21632\ln\left(\frac{6656\ln\left(\frac{208}{\tau}\right)}{\tau\beta_t}\right)}{\tau^2} \qquad (13)$$

$$N_t = ne^t \geq \frac{9984\ln\left(\frac{4M_t}{\beta_t}\right)\sqrt{32B_t\ln\left(\frac{1664\ln\left(\frac{208}{\tau}\right)}{\tau\beta_t}\right)}}{\tau^2} \qquad (14)$$

**Algorithm 5** EVERLASTINGTO$(\tau, \beta, p)$

---

1: Require sufficiently large initial budget $n$ (see proof of Theorem 6)
2: $\forall t$ set $N_t = ne^t$, $\beta_t = \frac{(e-1)\beta}{e}e^{-t}$, $B_t = \frac{\tau^4 N_t^{2-2p}}{8 \cdot 9984^2 \ln\left(\frac{1664 \ln\left(\frac{208}{\tau}\right)}{\tau \beta_t}\right)}$, $M_t = \frac{\beta_t}{4}\exp\left(2N_t^p\right)$

3: **for** $t = 0, 1, \ldots$ **do**
4:     Purchase datasets $S_t, T_t \sim \mathcal{D}^{N_t}$ and initialize THRESHOLDOUT$(S_t, T_t, B_t, \beta_t)$
5:     **while** THRESHOLDOUT$(S_t, T_t, B_t, \beta_t)$ has not halted **do**
6:        Accept query $q$
7:        $(a, o) =$ THRESHOLDOUT$(S_t, T_t, B_t, \beta_t)(q)$
8:        **Output**: $a$
9:        **if** $o = \perp$ **then**
10:           **Charge**: $\frac{2N_{t+1}}{M_t}$
11:        **else**
12:           **Charge**: $\frac{2N_{t+1}}{B_t}$

---

in order to satisfy the hypotheses of Theorem 5. Setting the constant $n$ such that

$$n \geq \frac{21632\left(1 + \ln\left(\frac{6656e\ln\left(\frac{208}{\tau}\right)}{(e-1)\tau\beta}\right)\right)}{\tau^2} \tag{15}$$

ensures that (13) holds. Furthermore, with our choice of

$$B_t = \frac{\tau^4 N_t^{2-2p}}{8 \cdot 9984^2 \ln\left(\frac{1664\ln\left(\frac{208}{\tau}\right)}{\tau\beta_t}\right)} \tag{16}$$

the condition (14) allows us to answer $M_t = \frac{\beta_t}{4}\exp\left(2N_t^p\right)$ total queries.

We also need to ensure that $1 \leq B_t \leq M_t$ $\forall t$ in order to ensure that THRESHOLDOUT has sound parameters. To satisfy $1 \leq B_t$ requires the initial budget $n$ to be sufficiently large as $p \to 1$.

$$1 \leq \frac{\tau^4\left(ne^t\right)^{2-2p}}{8 \cdot 9984^2 \ln\left(\frac{1664\ln\left(\frac{208}{\tau}\right)}{\tau\beta_t}\right)} \forall t \iff n \geq \sup_{t\in\mathbb{N}} e^{-t}\left(\frac{8 \cdot 9984^2\left(t + \ln\left(\frac{1664e\ln\left(\frac{208}{\tau}\right)}{(e-1)\tau\beta}\right)\right)}{\tau^4}\right)^{\frac{1}{2-2p}} \tag{17}$$

By Lemma 10, it thus suffices to choose

$$n \geq \left(\frac{8 \cdot 9984^2 \ln\left(\frac{1664e\ln\left(\frac{208}{\tau}\right)}{(e-1)\tau\beta}\right)}{\tau^4} + \frac{4 \cdot 9984^2}{(1-p)\tau^4}\right)^{\frac{1}{2-2p}} \tag{18}$$

At the same time, we need the initial budget to be large enough that $\forall t$ $B_t \leq M_t$:

$$M_t \geq B_t \qquad \forall t \tag{19}$$

$$\Longleftarrow \frac{(e-1)\beta}{4e}\exp\left(2n^p e^{pt} - t\right) \geq \frac{\tau^4\left(ne^t\right)^{2-2p}}{8 \cdot 9984^2 \ln\left(\frac{1664e\ln\left(\frac{208}{\tau}\right)}{(e-1)\tau\beta}\right)} \qquad \forall t \tag{20}$$

$$\iff \inf_{t\in\mathbb{N}} 2n^p e^{pt} - (3-2p)t - (2-2p)\ln n \geq \ln\left(\frac{e\tau^4}{2 \cdot 9984^2(e-1)\beta\ln\left(\frac{1664e\ln\left(\frac{208}{\tau}\right)}{(e-1)\tau\beta}\right)}\right) \tag{21}$$

By Lemma 11, the infimum can be lower bounded by $\ln n - \frac{3-2p}{p} \ln \frac{3-2p}{2ep}$ when $n \geq \left(\frac{3-2p}{2p}\right)^{1/p}$.
Therefore, $\forall t \; B_t \leq M_t$ is implied by

$$n \geq \max \left\{ \frac{e\tau^4 \left(\frac{3-2p}{2ep}\right)^{\frac{3-2p}{p}}}{2 \cdot 9984^2(e-1)\beta \ln\left(\frac{1664e\ln\left(\frac{208}{\tau}\right)}{(e-1)\tau\beta}\right)}, \left(\frac{3-2p}{2p}\right)^{1/p} \right\} \Longleftarrow n \geq \left(\frac{3-2p}{2p}\right)^{\frac{3-2p}{p}}$$

(22)

Therefore, in order to satisfy the hypotheses of Theorem 5, we require from (15), (18), and (22) that

$$n \geq \max \left\{ \frac{21632\ln\left(\frac{6656e^2\ln\left(\frac{208}{\tau}\right)}{(e-1)\tau\beta}\right)}{\tau^2}, \left(\frac{8 \cdot 9984^2\ln\left(\frac{1664e\ln\left(\frac{208}{\tau}\right)}{(e-1)\tau\beta}\right)}{\tau^4} + \frac{4\cdot 9984^2}{(1-p)\tau^4}\right)^{\frac{1}{2-2p}}, \left(\frac{3-2p}{2p}\right)^{\frac{3-2p}{p}} \right\}$$

(23)

Generally speaking, the first term will dominate when $p$ is relatively far from both zero and one, the second term will dominate as $p \to 1$, and the third term will dominate when $p \to 0$.

By Theorem 5, in round $t$, all answers returned by THRESHOLDOUT satisfy $|a_i - \mathbb{E}[q_i]| \leq \tau$ with probability $1 - \beta_t$. Therefore,

$$\mathbb{P}\left[\exists i \; |a_i - \mathbb{E}[q_i]| > \tau\right] \leq \sum_{t=0}^{\infty} \beta_t = \frac{(e-1)\beta}{e} \sum_{t=0}^{\infty} e^{-t} = \beta$$

(24)

$\square$

**Theorem 7.** *[Sustainability] For any $\tau, \beta, p \in (0,1)$ and any sequence of queries, EVERLASTINGTO charges enough for queries such that it can always afford to buy new datasets, excluding the initial budget.*

*Proof.* The $t^{\text{th}}$ instance of THRESHOLDOUT halts only after it has either answered $M_t$ total queries or at least $B_t$ queries with $o = \top$. In the first case, the total revenue is at least $M_t \cdot \frac{2N_{t+1}}{M_t} = 2N_{t+1}$ and in the latter case, the total revenue is at least $B_t \cdot \frac{2N_{t+1}}{B_t} = 2N_{t+1}$. Either way, it can affort to buy $S_{t+1}, T_{t+1}$, which have size $N_{t+1}$ each. $\square$

**Theorem 8.** *[Non-Adaptive Cost] For any $\tau, \beta, p \in (0,1)$, a sufficiently large initial budget, and any sequence of querying rules, the total cost, $\Pi$, to a non-adaptive user who makes $M$ queries to EVERLASTINGTO satisfies*

$$\mathbb{P}\left[\Pi > 2e^3 \ln^{1/p}\left(\frac{eM}{(e-1)\beta}\right)\right] \leq \beta$$

*Proof.* By Theorem 5's guarantee on THRESHOLDOUT and a union bound over all $t$, all non-adaptive queries are answered with $o = \bot$ with probability at least $1 - \sum_{t=0}^{\infty} \beta_t = 1 - \beta$. For the rest of the proof, we condition on this event.

First, observe that the cost of a query with $o = \bot$ is non-increasing over time, so the cost of any $M$ non-adaptive queries is no more than the cost of making the *first* $M$ non-adaptive queries. Let $T$ be the round in which the $M^{\text{th}}$ non-adaptive query is made if no adaptive queries are made.

Let $\Pi$ be the total amount paid. This is at most the total number of samples used in rounds 1 through $T+1$, i.e.

$$\Pi \leq \sum_{t=1}^{T+1} 2N_t = 2n \sum_{t=1}^{T+1} e^t \leq 2ne^{T+2}$$

(25)

Furthermore, the total number of queries made satisfies

$$M \geq M_{T-1} = \beta_{T-1} \exp\left(2N_{T-1}^p\right)$$

(26)

which implies

$$\ln\left(\frac{eM}{(e-1)\beta}\right) \geq 2N_{T-1}^p - (T-1) \geq N_{T-1}^p = n^p e^{p(T-1)} \tag{27}$$

where we use the fact that $n \geq (1/p)^{1/p}$ (see proof of Theorem 6) which implies $N_{T-1}^p = n^p e^{p(T-1)} \geq \frac{e^{p(T-1)}}{p} \geq \frac{p(T-1)}{p} = T-1$. Combining (25) and (27),

$$\Pi \leq 2ne^{T+2} \leq 2e^3 \ln^{1/p}\left(\frac{eM}{(e-1)\beta}\right) \tag{28}$$

$\square$

**Theorem 9.** *[Adaptive Cost] For any $\tau, \beta \in (0,1)$, $p \in (0, \frac{2}{3})$, a sufficiently large initial budget, and any sequence of querying rules, the total cost, $\Pi$, to a user who makes $B$ potentially adaptive queries to* EVERLASTINGTO *satisfies*

$$\mathbb{P}\left[\Pi \leq 2e^2\left(\frac{8 \cdot 9984^2 eB \ln\left(\frac{1664 \ln\left(\frac{208}{\tau}\right)}{(e-1)\tau\beta}\right)}{\tau^4}\right)^{\frac{1}{2-3p}}\right] = 1$$

*Proof.* First, observe that the cost of a query is non-increasing over time, so the cost of any $B$ adaptive queries is no more than the cost of making the *first* $B$ adaptive queries. Furthermore, adaptive queries may be answered with either $\top$ or $\bot$, but since $B_t \leq M_t$ $\forall t$, the cost of an adaptive query in round $t$ is no more than $\frac{2N_{t+1}}{B_t}$. Let $T$ be the round in which the $B^{\text{th}}$ adaptive query is made. Let $\Pi$ be the total amount paid. This is at most the total number of samples used in rounds 1 through $T+1$, i.e.

$$\Pi \leq \sum_{t=1}^{T+1} 2N_t = 2n \sum_{t=1}^{T+1} e^t \leq 2ne^{T+2} \tag{29}$$

Furthermore, the total number of adaptive queries is

$$B \geq \sum_{t=0}^{T-1} B_t = \sum_{t=0}^{T-1} \frac{\tau^4 N_t^{2-2p}}{8 \cdot 9984^2 \ln\left(\frac{1664 \ln\left(\frac{208}{\tau}\right)}{\tau\beta_t}\right)} \tag{30}$$

$$\geq \frac{\tau^4}{8 \cdot 9984^2 \left(T-1+\ln\left(\frac{1664e\ln\left(\frac{208}{\tau}\right)}{(e-1)\tau\beta}\right)\right)} \sum_{t=0}^{T-1} N_t^{2-2p} \tag{31}$$

$$= \frac{\tau^4 n^{2-2p}}{8 \cdot 9984^2 \left(T+\ln\left(\frac{1664\ln\left(\frac{208}{\tau}\right)}{(e-1)\tau\beta}\right)\right)} \sum_{t=0}^{T-1} e^{t(2-2p)} \tag{32}$$

$$\geq \frac{\tau^4 n^{2-2p}\left(e^{T(2-2p)}-1\right)}{8 \cdot 9984^2 T \ln\left(\frac{1664\ln\left(\frac{208}{\tau}\right)}{(e-1)\tau\beta}\right)} \tag{33}$$

$$\geq \frac{\tau^4 n^{2-2p} e^{T(2-2p)-1}}{8 \cdot 9984^2 T \ln\left(\frac{1664\ln\left(\frac{208}{\tau}\right)}{(e-1)\tau\beta}\right)} \tag{34}$$

Where in the last inequality we used that $p < \frac{2}{3}$ so $e^{T(2-2p)} - 1 \geq e^{T(2-2p)-1}$. Since $n \geq (1/p)^{1/p}$ (see proof of Theorem 6), it is also the case that $n^p e^{pT} \geq T$. Picking up from (34), we have

$$\frac{8 \cdot 9984^2 B \ln\left(\frac{1664\ln\left(\frac{208}{\tau}\right)}{(e-1)\tau\beta}\right)}{\tau^4} \geq \frac{n^{2-2p} e^{T(2-2p)-1}}{n^p e^{pT}} = n^{2-3p} e^{T(2-3p)-1} \tag{35}$$

thus

$$ne^T \le \left( \frac{8 \cdot 9984^2 eB \ln\left( \frac{1664 \ln\left(\frac{208}{\tau}\right)}{(e-1)\tau\beta} \right)}{\tau^4} \right)^{\frac{1}{2-3p}} \tag{36}$$

Combining (29) and (36), we get that

$$\Pi \le 2ne^{T+2} \le 2e^2 \left( \frac{8 \cdot 9984^2 eB \ln\left( \frac{1664 \ln\left(\frac{208}{\tau}\right)}{(e-1)\tau\beta} \right)}{\tau^4} \right)^{\frac{1}{2-3p}} \tag{37}$$

$\square$

To expand on the guarantees of Theorems 8 and 9, $p$ is a parameter of the algorithm that can be chosen roughly in the range $(0, 1)$. These theorems could be stated instead in terms of the quantity $a = 1/p$, which lies generally in the range $(1, \infty)$. In this case, a sequence of $M$ non-adaptive queries would cost (with high probability) at most $\mathcal{O}\left(\ln^a M\right)$, and a sequence of $M$ adaptive queries would cost at most $\mathcal{O}\left(B^{\frac{a}{2a-3}}\right)$. That is, when $a$ is near 1, we approach the optimal $\log M$ cost for non-adaptive queries at the expense of a very large (exploding) cost of adaptive queries. On the other hand, as we made $a$ very large, we approach the optimal $\sqrt{M}$ cost for adaptive queries at the expense of more expensive polylog cost for non-adaptive queries. In this way, the parameter $p$ trades off between placing the burden of adaptivity directly on the adaptive queries themselves and spreading it out over potentially non-adaptive queries too.

**Lemma 10.** *For any $\beta, \tau, p \in (0, 1)$,*

$$\sup_{t \in \mathbb{N}} e^{-t} \left( \frac{8 \cdot 9984^2 \left( t + \ln\left( \frac{1664e \ln\left(\frac{208}{\tau}\right)}{(e-1)\tau\beta} \right) \right)}{\tau^4} \right)^{\frac{1}{2-2p}} \le \left( \frac{8 \cdot 9984^2}{\tau^4} \left( \ln\left( \frac{1664e \ln\left(\frac{208}{\tau}\right)}{(e-1)\tau\beta} \right) + \frac{1}{2-2p} \right) \right)^{\frac{1}{2-2p}}$$

*Proof.* For brevity, let $a := \frac{8 \cdot 9984^2}{\tau^4}$, let $b := \ln\left( \frac{1664e \ln\left(\frac{208}{\tau}\right)}{(e-1)\tau\beta} \right)$, and let $c = \frac{1}{2-2p}$, note that $a, b, c > 0$. We are thus interested in upper bounding $\sup_{t \in \mathbb{N}} e^{-t} \left( at + ab \right)^c$. First,

$$\frac{d}{dt} e^{-t} \left( at + ab \right)^c = ace^{-t} \left( at + ab \right)^{c-1} - e^{-t} \left( at + ab \right)^c \tag{38}$$

and

$$ace^{-t} \left( at + ab \right)^{c-1} - e^{-t} \left( at + ab \right)^c = 0 \iff t = c - b \text{ or } t = -b \text{ or } t \to \infty \tag{39}$$

Since we are only optimizing over $t \in \mathbb{N}$ and $b > 0$, we do not need to consider the critical point $t = -b$. Furthermore,

$$\left. \frac{d^2}{dt^2} e^{-t} \left( at + ab \right)^c \right|_{t=c-b} = -\frac{1}{c} (ac)^c e^{b-c} < 0 \tag{40}$$

Therefore, the critical point at $t = c - b$ is a local maximum. Therefore, the only points we need to consider are when $t = 0$, $t \to \infty$, and $t = c - b$ if $c \ge b$.

$$\sup_{t \in \mathbb{N}} e^{-t} \left( at + ab \right)^c \le \begin{cases} (ab)^c & b > c \\ \max\left\{ (ab)^c, e^{b-c}(ac)^c \right\} & c \ge b \end{cases} \le a^c (b + c)^c \tag{41}$$

which completes the proof. $\square$

**Lemma 11.** *For any $p \in (0, 1)$ and $n \ge 1$*

$$\inf_{t \in \mathbb{N}} 2n^p e^{pt} - (3 - 2p)t - (2 - 2p) \ln n \ge \min\left\{ \ln n - \frac{3 - 2p}{p} \ln \frac{3 - 2p}{2ep}, \ 2n^p - (2 - 2p) \ln n \right\}$$

*and the first term is the minimizer when $n \ge \left( \frac{3-2p}{2p} \right)^{1/p}$*

*Proof.* First, note that this is a convex function in $t$ and

$$\frac{d}{dt} 2n^p e^{pt} - (3 - 2p)t - (2 - 2p) \ln n = 2pn^p e^{pt} - 3 + 2p \tag{42}$$

and

$$2pn^p e^{pt} - 3 + 2p = 0 \iff t = \frac{1}{p} \ln \frac{3 - 2p}{2p} - \ln n \tag{43}$$

Therefore, if $\frac{1}{p} \ln \frac{3-2p}{2p} - \ln n \geq 0$ then

$$\inf_{t \in \mathbb{N}} 2n^p e^{pt} - (3 - 2p)t - (2 - 2p) \ln n \geq \ln n - \frac{3 - 2p}{p} \ln \frac{3 - 2p}{2ep} \tag{44}$$

Otherwise, if $\frac{1}{p} \ln \frac{3-2p}{2p} - \ln n < 0$

$$\inf_{t \in \mathbb{N}} 2n^p e^{pt} - (3 - 2p)t - (2 - 2p) \ln n \geq 2n^p - (2 - 2p) \ln n \tag{45}$$

Thus,

$$\inf_{t \in \mathbb{N}} 2n^p e^{pt} - (3 - 2p)t - (2 - 2p) \ln n \geq \min \left\{ \ln n - \frac{3 - 2p}{p} \ln \frac{3 - 2p}{2ep}, \ 2n^p - (2 - 2p) \ln n \right\} \tag{46}$$

$\square$

## E   Relevant Results in Differential Privacy

Here, we state without proof definitions and results from other work which we use in the proof of Lemma 6.

**Definition 1.** *A randomized algorithm* $\mathcal{M} : \mathcal{X}^* \mapsto \mathcal{Y}$ *is* $(\epsilon, \delta)$-*differentially private if for all* $E \subseteq \mathcal{Y}$ *and all datasets* $S, S' \in \mathcal{X}^*$ *differing in a single element:*

$$\mathbb{P}\left[\mathcal{M}(S) \in E\right] \leq e^\epsilon \mathbb{P}\left[\mathcal{M}(S') \in E\right] + \delta.$$

**Proposition 1** ([4, 18]). *Let* $\mathcal{M}$ *be an* $(\epsilon, \delta)$-*differentially private algorithm that outputs a function from* $\mathcal{X}$ *to* $[0, 1]$*. For a random variable* $S \sim \mathcal{D}^n$ *we let* $q = \mathcal{M}(S)$*. Then for* $n \geq 2 \ln(8/\delta)/\epsilon^2$,

$$\mathbb{P}\left[|\mathcal{E}_S[q] - \mathbb{E}[q]| \geq 13\epsilon\right] \leq \frac{2\delta}{\epsilon} \ln \left(\frac{2}{\epsilon}\right).$$

**Definition 2** (Definition 1.1 [6]). *A randomized mechanism* $M : \mathcal{X}^n \to \mathcal{Y}$ *is* $\rho$-*zero-concentrated differentially private (henceforth* $\rho$-*zCDP) if, for all* $S, S' \in \mathcal{X}^n$ *differing on a single entry and all* $\alpha \in (1, \infty)$,

$$D_\alpha\left(\mathcal{M}(S) || \mathcal{M}(S')\right) \leq \rho\alpha,$$

*where* $D_\alpha\left(\mathcal{M}(S) || \mathcal{M}(S')\right)$ *is the* $\alpha$-*Rényi divergence between the distribution of* $\mathcal{M}(S)$ *and* $\mathcal{M}(S')$.

**Proposition 2** (Proposition 1.6 [6]). *Let* $q$ *be a statistical query. Consider the mechanism* $\mathcal{M} : \mathcal{X}^n \to \mathbb{R}$ *that on input* $S$*, releases a sample from* $\mathcal{N}(\mathcal{E}_S[q], \sigma^2)$*. Then* $\mathcal{M}$ *satisfies* $\frac{1}{2n^2\sigma^2}$-*zCDP.*

**Proposition 3** (Lemma 1.7 [6]). *Let* $\mathcal{M} : \mathcal{X}^n \to \mathcal{Y}$ *and* $\mathcal{M}' : \mathcal{X}^n \to \mathcal{Z}$ *be randomized algorithms. Suppose* $\mathcal{M}$ *satisfies* $\rho$-*zCDP and* $\mathcal{M}'$ *satisfies* $\rho'$-*zCDP. Define* $\mathcal{M}'' : \mathcal{X}^n \to \mathcal{Y} \times \mathcal{Z}$ *by* $\mathcal{M}''(x) = (\mathcal{M}(x), \mathcal{M}'(x))$*. Then* $\mathcal{M}''$ *satisfies* $(\rho + \rho')$-*zCDP.*

**Proposition 4** (Proposition 1.3 [6]). *If* $\mathcal{M}$ *provides* $\rho$-*zCDP, then* $\mathcal{M}$ *is* $\left(\rho + 2\sqrt{\rho \ln(1/\delta)}, \delta\right)$-*differentially private for any* $\delta > 0$.