[Reviews · NeurIPS 2018]

Reviewer 1



This paper introduces a model for “fairly” pricing (statistical) queries to a database in a model where there are two types of users (adaptive and non-adaptive) and the database can purchase new data at a price. This price is charged differently to the two types of users, leading to a price per-query of (1/sqrt(M)) or (log M/ M). My feeling about this paper is that it sets up a rather complicated database access and pricing model, which then is shown to be feasible via known differential privacy / adaptivity, validity, and simple pricing results by adding the right amount of noice, checking answers, etc. I do not see much of an advance in any fundamental problem, but rather a rather complicated model and results to fit it. The theorems are quite believable, though I did not check the appendix. The paper is well written. Unfortunately, I am simply not convinced by the results themselves and therefore think this paper is below the bar for acceptance. One note: if the authors pointed to the need for such systems, or better yet to current attempts to build them, in practice, I would be more optimistic about this line of work. minor comments Fairness is mentioned in the title, but nowhere else in the paper. I don’t think that fairness is the best descriptor of what you’re doing.

Reviewer 2



The paper studies the problem of “adaptive data analysis” and proposed a new model where the data base can gather new data at a price, and also users of this data base need to pay to use it. The proposed model is appealing in that it does not model users at all, yet it can achieve the desired behavior w.r.t. each user regardless of whether the user is adaptive or not. The main contribution of this paper is the above new setting and an analysis of the proposed algorithm. Pros: - The problem is important. The setting is practically relevant. - The results seem technically correct. - The discussions and future work are well-thought out. I particularly would love to see the “typical charges” in practice. Cons: - The technical novelty is on the low end. The authors are pretty frank about this too, so it is perhaps not fair to press on this matter too much. - For NIPS (rather than COLT), I feel that the authors should have a careful simulation study to illustrate the behaviors of different kind of users. - The model is still rather idealized for practical use. The pricing is a bit extreme. People are penalized for providing something that produces inconsistent results on the two sets, which could be due to pure noise especially when \tau is small.

Reviewer 3



This paper studies the problem of answering a (possibly infinite) sequence of (adaptive and non-adaptive) statistical queries without overfitting. Queries trigger the acquisition of fresh data when the mechanism determines that overfitting is likely, so adaptive queries necessitate new data. By continually acquiring fresh data as needed, the mechanism can (whp) guarantee accuracy in perpetuity. Moreover, by passing on the "cost" of data acquisition to queries that trigger it, the mechanism guarantees that (whp) non-adaptive queries pay cost O(log(# queries)) while adaptive queries pay cost O(sqrt(# queries)). Suggested applications are the normal ones for adaptive data analysis: ML competition leaderboards and scientific discovery. ---Positive points (no order)--- 1. This paper is clear throughout: the introduction sets up the (natural) problem well, and the descriptions of the mechanisms and their guarantees are easily understandable. The comparison to known work is reasonably comprehensive while remaining brief, as is the sketch of optimality. Reading this paper is pleasant. This is partly due to quality writing and partly due to the relatively simple (or, more charitably, elegant) ideas used. 2. The varying cost guarantees for adaptive and non-adaptive users, arising as a natural outgrowth of being adaptive or non-adaptive, are nice. 3. As somebody who's generally more familiar with the differential privacy literature than the adaptive data analysis literature, I'm happy to see concentrated differential privacy working its way into adaptive data analysis -- as far as I know this is the first adaptive data analysis paper to use these tools. ---Negative points (in order of decreasing importance)--- 1. I think it ultimately falls on the right side of this divide, but this paper is close to being "too" simple. By my count there is only one result (Lemma 6) in this paper that takes more than a minute or so to grok, and I’m by no means an expert in either differential privacy or adaptive data analysis. Indeed the protocol as a whole is very close to well-known basic adaptive data analysis protocols. The addition of data acquisition is a nice wrinkle, but the result feels a bit like the first naive protocol someone familiar with adaptive data analysis would provide if given this problem. It's silly to say "I am rejecting this paper because it is not long or hard enough!", but the paper does feel "thin" and even a little obvious. Of course, this obviousness may only be in retrospect, but I feel this is a paper that could move from "marginal accept" to "accept" with just a little more work. Even just writing up the "more formal treatment" of the cost-spreading mechanism mentioned in the conclusion, shaving down the cost guarantee constants, or incorporating growing data acquisition costs (which feels a little more realistic) would suffice. 2. The use of "minus current capital" in the EverlastingValidation pseudocode is somewhat vague, it would make more sense to me to actually write this as a variable that's modified with charges. ---Overall--- This paper does a good job of introducing its problem, outlining a clear solution and analysis, and situating its contribution in the existing literature. The same simplicity that makes for enjoyably clear reading also makes it hard to wholeheartedly recommend this paper for acceptance: while reading, my immediate reactions were primarily “yup, that’s what you’d do there” rather than “oh, I see, I now understand that's unfamiliar technique”. As a non-expert in both differential privacy and adaptive data analysis, this gives me pause even if I like the paper – I’m just not sure it’s making a strong contribution to the state of the art. Again, I really wish the authors had just chased down one of the extensions they mention! As a result, I marginally recommend acceptance. ---Update After Response--- After reading the authors' response, I am increasing my score from a 6 to a 7. My broad perception of the paper is essentially the same: I think it's a useful result, but a little thin as a NIPS paper. I'm willing to defend the paper a bit.